# Effect of Y Addition on the Microstructure and Mechanical Properties of ZM31 Alloy

**DOI:** 10.3390/ma13030583

**Published:** 2020-01-26

**Authors:** Xue Ye, Hongshuai Cao, Fugang Qi, Xiaoping Ouyang, Zhisong Ye, Caihong Hou, Lianhui Li, Dingfei Zhang, Nie Zhao

**Affiliations:** 1School of Materials Science and Engineering, Xiangtan University, Xiangtan, 411105, China; yexue2201@163.com (X.Y.); caohongshuai@aliyun.com (H.C.); yezhisong@163.com (Z.Y.); houcaihong19@163.com (C.H.); lilianhui_123@163.com (L.L.); zhaonie@xtu.edu.cn (N.Z.); 2College of Materials Science and Engineering, Chongqing University, Chongqing, 400045, China; zhangdingfei@cqu.edu.cn

**Keywords:** Mg-3Zn-1Mn alloy, α-Mn, Y content, W-phase, LPSO phase

## Abstract

Effects of different Y contents (0, 0.3, 0.7, 1.5, 3, 5 and 10 wt.%) on the microstructure, thermal stability and mechanical properties of Mg-3Zn-1Mn (ZM31) alloys were systematically studied. The existence form and action mechanism of Y in the experimental alloys were investigated. The results revealed that with the change of Y content, the main phases of the ZM31-xY alloys changed from Mg_7_Zn_3_ phase, I-phase, I + W-phase, W-phase, W + LPSO phase to LPSO phase. When Y content was low (≤1.5%), hot extrusion could break up the residual phases after homogenization to form dispersed fine rare-earth phase particles, and fine second phases would also precipitate in the grain, which could inhibit the grain growth. When Y content was high (≥3%), the experimental alloys were only suitable for high-temperature extrusion due to the formation of the high heat stable rare-earth LPSO phase. In addition, Y could evidently enhance the mechanical properties of the as-extruded ZM31 alloy, of which the ZM31-10Y alloy had the best mechanical properties, that is, the tensile and yield strengths are 403 MPa and 342 MPa. The high strengths of the alloys were mainly determined by fine grain strengthening, rare-earth phase strengthening and dispersion strengthening of fine α-Mn particles.

## 1. Introduction

Magnesium (Mg) alloys are currently the lightest metal structural material in the industry, which are widely used in the transportation, electronics and military industries [1,2,3]. At present, the application of Mg alloys is still largely inferior to Al alloys. The main reasons are that the absolute strength of Mg alloys is still low, and the plastic deformation processing ability at room temperature is poor. [4,5,6]. Therefore, many researchers have focused their research on the development of high-strength wrought Mg alloys. Alloying, especially the addition of rare-earth (RE, mainly including Y, Gd, Nd, Ce, Er and Sr) elements to Mg alloys, is known to be an effective method to decrease the influence of temperature and obtain alloys with high strength and ductility [5,7].

At present, the Mg-Zn-RE series alloy has become one of the most important alloy systems due to its excellent room temperature and high-temperature mechanical properties, and unique microstructures, which has attracted the increasing attention of researchers [8]. Among the RE elements, Y has attracted widespread attention [9]. It has been reported that Y element can effectively enhance the mechanical properties, high-temperature creep resistance and corrosion resistance of Mg-Zn series alloys [10,11]. The excellent mechanical properties of Mg-Zn-Y series alloys are mainly attributed to three types of RE-containing phases, namely I-phase (Mg_3_Zn_6_Y, icosahedral quasicrystal), W-phase (Mg_3_Zn_3_Y_2_) and long-period stacking-ordered (LPSO, Mg_12_Zn_1_Y_1_) phase. Earlier studies have reported that I-phase has a three-dimensional quasiperiodic structure, which has particular properties such as high hardness, coherence with the matrix and low surface energy [12]. Researchers have found that W-phase and LPSO phase can also significantly improve the mechanical properties of Mg alloys [13,14,15,16]. In a recent study, Wang et al. found that W-phase and LPSO phase may play a composite strengthening role together [17]. The formation of these second phases depends not only on the production process, but also on the atomic ratio of Zn and Y in particular atomic planes. For researchers, it has significant meaning to further clarify the second phase structure and microstructure evolution by changing the Zn/Y ratio. However, the current research on the evolutionary behavior of these RE-containing phases and how they affect the mechanical properties of the RE Mg alloys is not systematic. Hence, it is very necessary to determine the effects of these RE-containing phases and the influence of their evolution on the properties of the Mg-Zn-Y series alloys. 

Moreover, by adding Mn, the grain size of the Mg alloy is refined, and the mechanical properties are significantly improved [18]. In recent years, researchers have discovered that Mn addition to Mg-Zn alloys can refine the grains and significantly improve the mechanical properties of the alloys. Based on this, a new type of Mg-Zn-Mn wrought Mg alloy has been developed. The Mn in the Mg-Zn-Mn alloy mainly plays a major role of grain refinement, which is similar to the role of Zr in commercial Mg-Zn-Zr alloys [19,20]. Therefore, the influence of different Y contents (0, 0.3, 0.7, 1.5, 3, 5 and 10 wt.%) on the microstructure, thermal stability and mechanical properties of the as-cast, as-homogenized and as-extruded Mg-3Zn-1Mn (ZM31) alloys is systematically studied. 

## 2. Experimental Details

ZM31-based alloys with different Y contents (0%, 0.3%, 0.7%, 1.5%, 3%, 5% and 10%) were prepared. The raw materials mainly included industrial pure Mg, pure Zn, Mg-30.29%Y and Mg-4.1% Mn master alloys, all of which have a purity of 99.9%. All the added materials were melted together in a steel crucible in a resistance furnace under the protection of Ar to prevent oxidation, and then cast into a steel mold. Then, a XRF-1800CCDE X-ray fluorescence spectrometer (XRF) was used to test the composition of the experimental alloy ingots. The test sample was a cylindrical sample with a diameter between 30 and 48 mm. The scanning method was performed using the "area scanning" method. The actual composition is shown in Table 1. Overall, the actual chemical compositions of all alloy ingots met the previous design requirements. Then, different homogenization processes were performed on the investigated alloy ingots with different Y contents, that is, ZM31 and ZM31-0.3Y alloys: 420 °C/12 h, and all other alloys: 500 °C/16 h, followed by air cooling. Then, the as-homogenized alloy ingots were extruded on an XJ-500 horizontal extrusion machine, and the extrusion process parameters are shown in Table 2. 

The extruded bars were tested for tensile properties at room temperature. According to GB228-2002, the tensile sample was designed as a cylinder with a gauge length of 60 mm and a diameter of the test rod within the gauge length of 5 mm. The mechanical performance test was performed on a SANS CMT-5105 microcomputer controlled electronic universal testing machine. Uniform speed unidirectional displacement stretching was used, and the stretching rate was 3 mm/s. Five sets of experiments were performed for each alloy bar. From the obtained stress–strain curve, the mechanical properties of the alloys are determined, that is, 0.2% yield strength (YS), ultimate tensile strength (UTS) and elongation to failure (elongation).

The D/MAX-2500PC X-ray diffractometer (XRD) was used for phase analysis of the test alloys. The scanning angle was 10°–90°, the scanning speed was 4°/min, and a Cu target was used. Olympus BX53M was used for metallographic analysis of the test alloys. The secondary electron (SE) and backscattered electron (BSE) probes of the scanning electron microscope (SEM) were used to observe the morphology of the corresponding samples in combination with EDS, and the second phase was analyzed semi-quantitatively. The SEM model used was TESCAN VEGA IIL, and the EDS model was OXFORD INCA Energy 350. Transmission electron microscope (TEM) observations were performed on a FEI Titan G2 Model 60–300. The NETZSCH STA 449C differential scanning calorimeter (DSC) was used for thermal analysis of the test alloy, and the test was performed under the protection of high-purity Ar gas flow. The temperature was raised from 50 °C to 700 °C at a heating rate of 10 K/min, and the temperature was maintained for 5 minutes, and then reduced to 50 °C at a cooling rate of 10 K/min. 

## 3. Results and Discussion

### 3.1. As-Cast and as-Homogenized Microstructures

Figure 1 shows the optical micrographs of the as-cast ZM31-xY alloys. It is found that the as-cast microstructures are mainly made up of α-Mg matrix and a mass of intermetallic compounds, which are distributed along the dendritic boundaries. Compared with Y-free ZM31 alloy, the addition of Y element results in significant changes in the microstructure. When the Y content is 0.3%, the eutectic compounds increase and the dendrites are refined. When the Y content reaches 0.7%, the fish bone network compounds begin to appear. When the Y content increases to 1.5%, all the eutectic compounds appear as the fish bone networks. When the Y content is 3%, new lamellar structure compounds appear in the matrix. When the Y content is 5% and 10%, all the fish bone eutectic compounds are replaced by the lamellar shape and dendrites are further refined. Generally speaking, adding Y element is useful to decrease the dendrite size, change microstructure and influence volume fraction of the second phases.

To further study the phase transformation of the test alloys, the phases of the as-cast ZM31-xY alloys are analyzed by XRD (Figure 2) and corresponding phase analysis results are summarized in Table 3. The phase composition of ZM31-xY alloys mainly includes α-Mg, Mn, Mg_7_Zn_3_, I-phase, W-phase and LPSO phase. As indicated in Figure 2, it is easy to see that second phases change with the Zn/Y mole ratio in the as-cast test alloys. According to the analysis results of the optical microstructure, it can be preliminary inferred that the eutectic compound in the ZM31 alloy is the Mg_7_Zn_3_ phase, the second phases in ZM31-0.3Y alloy are the mixture of Mg_7_Zn_3_ and I-phase, the fish bone network phase is a W-phase and the phase with lamellar structure is the LPSO phase.

Figure 3 reveals the typical SEM images of the as-cast ZM31-xY alloys. It is further confirmed that with the increase of the Y content, the volume fraction of the second phases increase and the dendritic structure is refined. From ZM31 alloy to ZM31-10Y alloy, the second phase structures changes from a gray white coarse structure (marked as B), fine strip-like structure (marked as C and D), and network structure (marked as E) to a gray block structure (marked as F). Combining XRD (Figure 2) and EDS analysis results (Table 4), it can be concluded that A is α-Mg matrix, B with a gray white coarse structure is identified as Mg_7_Zn_3_, C and D are I-phase, E shown in network structure is W-phase and F is LPSO phase. In addition, there are some bright white granular particles for the ZM31-10Y alloy in Figure 3g, which have been confirmed as RE-rich particles in many other studies [21,22].

Figure 4 reveals the DSC curves of the as-cast ZM31-xY alloys. The important temperature parameters extracted from the figure are summarized in Table 5. Three endothermic peaks are observed on the curve, and their temperatures are approximately 525 °C (peak1), 540 °C (peak 2) and 620 °C (peak3), respectively. According to the Mg-Zn-Y ternary phase diagram and the analysis results of previous studies [23,24], we can infer that the temperature of peak 1 is related to the dissolution of the W-phase, the temperature of peak 2 is related to the dissolution of the LPSO phase and the temperature of peak 3 is related to the melting of the test alloys. By comparing peak 2 and peak 3, it can be seen that the LPSO phase has a higher dissolution temperature, which indicates that it may have better thermal stability than the W-phase. With the increasing Y content, peak 3 is shown to decrease evidently, which indicates that the formation of RE phase is conducive to improve the castability of the alloys. In addition, no obvious peak for I-phase transformation is found in Figure 4, which may be due to too little I-phase being detected. 

It is worth noting that W-phase observed in the as-cast investigated alloys is a network structure. From previous research, Xu et al. has found that the formation of network W-phase will result in the deterioration of mechanical properties of the Mg-Zn-Y alloy [25]. Therefore, for the purpose of reducing the adverse effect of the network-like W-phase, the W-phase is dissolved as much as possible during the homogenization treatment. According to the DSC results, the melting temperatures of the W-phase and LPSO phase are both higher than 500 °C, which indicates that for the ZM31- (0.7–10) Y alloys with W-phase and LPSO phase, the homogenization treatment temperature is predicted to be higher than that of the ZM31- (0–0.3) Y alloys with only the Mg_7_Zn_3_ phase and I-phase. Therefore, ZM31-1.5Y and ZM31-3Y alloys are used as research objects, and two homogenization treatment temperatures of 420 °C/12 h and 500 °C/16 h are performed.

Figure 5 shows the SEM images of as-homogenized ZM31-1.5Y and ZM31-3Y alloys under different treatment temperatures. For the ZM31-1.5Y alloy, the second phase is mainly composed of the W-phase. Comparing Figure 5a,b, it is easy to find that the W-phase still maintains a network structure after heat treatment at 420 °C, while the W-phase is significantly dissolved and diffused after homogenization treatment at 500 °C. For the ZM31-3Y alloy, the second phase mainly consists of the W-phase and LPSO phase. Comparing Figure 5c,d, the W-phase has the same experimental results, but the structure of the LPSO phase has not changed even at 500 °C homogenization treatment. This result provides a good reference for the design of the homogenization process for the test alloys, which means that a higher homogenization temperature should be used for the alloys containing W-phase. In this way, it is possible to better transform the network-like W-phase to the particle phase during the extrusion process, and to achieve a dispersion strengthening effect. Therefore, different homogenization processes are performed for the test alloys containing different Y contents, that is, ZM31 and ZM31-0.3Y alloys: 420 °C/12 h, other alloys: 500 °C/16 h.

Figure 6 exhibits the optical images of the as-homogenized ZM31-xY alloys. Compared with Figure 1, the volume fraction of the residual phases after the homogenization treatment is significantly reduced, and the distribution of the second phases becomes thinner and discontinuous. Figure 7 presents the SEM images of the as-homogenized ZM31-xY alloys, and Table 6 lists the corresponding EDS results for second phases. According to EDS results, A, B, C, D, E, F and G are identified as α-Mg matrix, I-phase, W-phase, W-phase, LPSO phase, Y-rich phase and LPSO phase, respectively. Compared with the SEM results of the as-cast alloys (Figure 3), it can be found that the volume fraction of the Mg_7_Zn_3_ phase decreases dramatically, the I-phase becomes discontinuous and the W-phase also transforms from the network structure to a discontinuous honeycomb-like network. However, the LPSO phase still remains the block structure, and the result is the same as that shown in Figure 5.

In addition to the morphology of the second phases, there is an interesting phenomenon that as the Y content increases, many cuboid-shape Y-rich partials still remain in alloys and precipitate further. The possible reason is that Y content is already much larger than the required value. From the previously analyzed SEM results (Figure 3e–g) and the XRD results of as-homogenized ZM31-xY alloys containing the LPSO phase (Figure 8), it can be seen that as the Y content increases, the volume fraction of LPSO phase in the experimental alloys increases. When the Y content is equal to 5%, the Zn/Y ratio is lower than the transition threshold of the RE phase, which hinders the further growth of the LPSO phase. As can be clearly seen in Figure 7g, compared to the as-extruded ZM31-5Y alloy, the number of bright white square particles in the as-extruded ZM31-10Y alloy increases.

### 3.2. As-Extruded Microstructure

Figure 9 reveals the SEM images of the as-extruded ZM31-xY alloys, the observation direction is parallel to the extrusion direction. After the hot extrusion, the second phases are broken and distributed at the grain boundary. In Figure 9a, the grain size of the as-extruded ZM31 alloy has changed greatly after the extrusion due to the occurrence of the dynamic recrystallization (DRX) during the hot extrusion. When the content of Y is equal to 0.3%, 0.7% and 1.5%, the average grain size of the alloys decreases to about 4.5 μm, 2.5 μm and 1.5 μm, respectively. Moreover, it is also clear that there are many small bright white particles parallel to the extrusion direction, which are the second phases that we have been observed before (I-phase and eutectic W-phase). During the homogenization treatment, the I-phase and W-phase are effectively decomposed and dissolved, so for the alloys with a Y content not higher than 1.5%, they can be successfully extruded at low temperatures (350 °C) However, with the increase of the Y content, the LPSO phase is formed, and it is difficult for the alloys to be successfully extruded at low temperatures, and higher temperatures (480 °C) are required for extrusion processing. From Figure 9e–f, it can be seen that the grain size of the alloys with the high Y content (≥3%) becomes larger than that of the low Y content (≤1.5%), and the average grain size is increased to 5 μm, 8.3 μm and 10.5 μm, respectively. There are mainly two reasons, one is that the appearance of the LPSO phase accelerates the DRX, and the other is that the alloys containing the LPSO phase require a longer extrusion time during the process, providing the grain more time to develop. In the previous studies, it has been reported that the existence of the LPSO phase strongly accelerates the refinement of recrystallized grains of the alloy during the extrusion at higher temperature, and the LPSO phase itself also acts directly as a strong reinforcement in the alloy resulting in higher extrusion temperature [26,27]. Figure 9e–f shows the microstructure of the alloys extruded at a higher temperature of 480 °C, the microstructure of the LPSO phase does not change much after the extrusion process, and still distributes at the grain boundary with the grey block, which is consistence with the results of Figure 5.

To further study the second phase in the test alloys, TEM microstructure observation is carried out on the extruded ZM31-3Y alloy. Figure 10a shows a bright-field (BF) TEM micrograph of the second phase, which is identified as the W-phase with an FCC structure by the SAED pattern. In Figure 9b, it can be seen that a lot of gray second phases have a different direction. These phases are LPSO phases with a crystal distance of 1.8 nm, which are consistent with the results of the LPSO phase in 14H reported in the literature. Many studies have reported that 18H is unstable and will change to 14H at 480 °C [25,28,29]. Hence, it can be inferred that the LPSO phase is 14H after high-temperature homogenization and extrusion. In addition to the second phase, many black particle phases are dispersed in the alloy matrix, as shown in Figure 10d. According to the results of EDX analysis (Figure 10e–f), the black particle phase are the pure Mn phases. It is well known that Mn addition can significantly refine the grain size of the Mg alloy, and it is dispersed in the matrix as a dispersed particle phase, which helps to improve the mechanical properties of the alloy [19,20]. In this work, the Mn phase has been further evidenced in this form.

### 3.3. Mechanical Properties

Uniaxial tensile experiment is used to test the room temperature mechanical properties of the as-extruded experimental alloys. The results are shown in Figure 11. In this work, the relationship between the mechanical properties and the second phase will be discussed. Researchers have found that the appearance of the LPSO phase significantly improves the mechanical properties of the alloys, but low-temperature extrusion is difficult for such alloys, and high-temperature extrusion is generally used. Therefore, according to the characteristics of different RE phases, for alloys with different Y contents, different temperatures are selected for hot extrusion. When the Y content is less than 3%, the alloy extrusion temperature is 350 °C; when the Y content is 3% or more, the extrusion temperature is 480 °C. When the Y content is less than 3% and the extrusion temperature is 350 °C, as the Y content increases, the mechanical properties of the experimental alloys generally increase, that is, the ultimate tensile stress (UTS) increases from 275 MPa to 334 MPa and the yield tensile stress (YTS) increased from 207 MPa to 324 MPa. The mechanical properties may be determined by two factors: the appearance of second phases and the refined grain size. However, when the Y content is 0.7% and 1.5%, the grain size of the extruded alloys is fine and the mechanical properties are not significantly improved. This may be due to the W-phase. In this work, it has been further evidenced that the W-phase is not good at improving the strength of the experimental alloys. When the Y content is 3% or more, the LPSO phase begins to appear. The extrusion temperature is increased from 350 °C to 480 °C. It is found that with increasing Y content, the volume fraction of the LPSO phase increases, and the strengths of the extruded alloys increase. When the Y content increases from 3% to 10%, the Zn/Y molar ratio changes from 1.27 to 0.4, UTS increases from 285 MPa to 403 MPa, and YTS increases from 218 MPa to 342 MPa. Comparing ZM31-3Y with ZM31 under 350 °C extrusion, there is no significant change in performance, but from the extrusion process, it can be understood that the alloys containing LPSO phase should have better strength. This is because the appearance of the LPSO phase slows down the extrusion speed and is not even extruded until the temperature increases. In addition to the increase of strength, the elongation has decreased dramatically. It is indicated that the alloys with high strength will have poor flexibility in the process. Therefore, how to get the balance of strength and flexibility is the key to result in the wider use of the magnesium alloy. From these results, it can be seen that LPSO is more effective in enhancing alloy strength than the I- and W-phases. 

## 4. Conclusions

The Mg-3Zn-1Mn–xY (x = 0%, 0.3%, 0.7%, 1.5%, 3%, 5% and 10%) experimental alloys have been prepared by smelting, casting, homogenizing treatment and hot extrusion. The microstructure evolution and mechanical properties of the experimental alloys have been systematically studied by means of XRD, OM, SEM, TEM, DSC and tensile tests. The obtained results are summarized as follows:The Y addition has an obvious effect on the phase composition and microstructure of the ZM31 alloy. On one hand, the Y addition can significantly refine the dendrite size of the as-cast ZM31 alloy, and on the other hand, various phase compositions including Mg_7_Zn_3_, I-phase, W-phase, and LPSO phase can be obtained by adjusting the Zn/Y ratio. Based on the results of thermal analysis and microstructural observations, the phase stability follows the trend of LPSO phase> W-phase> I-phase> Mg_7_Zn_3_ phase.Y can significantly improve the room temperature mechanical properties of the as-extruded ZM31 alloy. Under the same extrusion conditions, as the Y content increases, the mechanical properties show an increasing trend. Among them, the ZM31-10Y alloy with LPSO phase has the best mechanical properties, that is, the UTS and YTS reach 403 MPa and 342 MPa, respectively.

## Figures and Tables

**Figure 1 materials-13-00583-f001:**
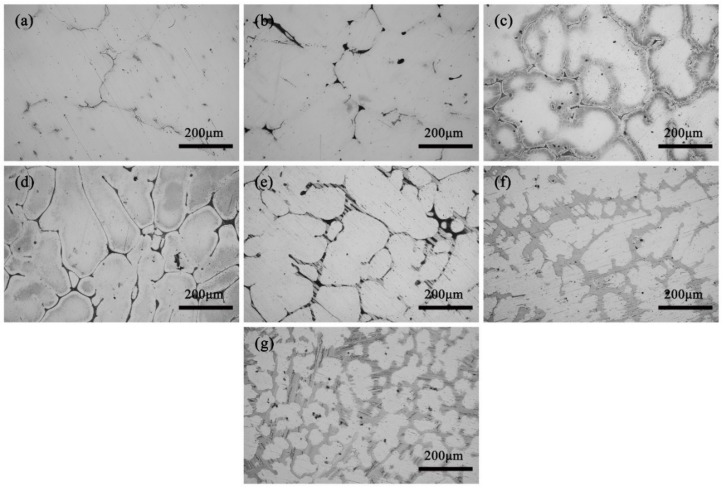
Optical micrographs of the as-cast ZM31-xY alloys: (**a**) x = 0; (**b**) x = 0.3; (**c**) x = 0.7; (**d**) x = 1.5; (**e**) x = 3; (**f**) x = 5; (**g**) x = 10.

**Figure 2 materials-13-00583-f002:**
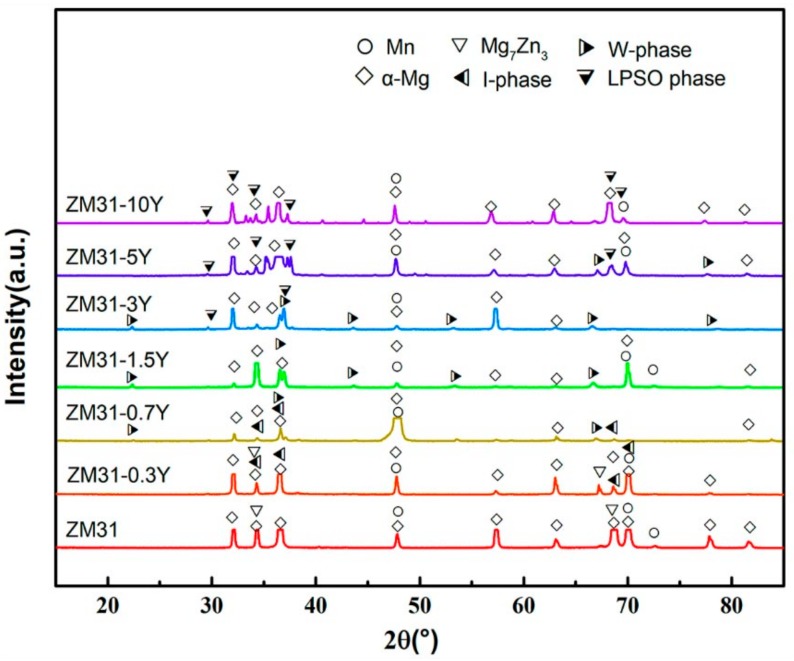
X-ray diffractometer (XRD) analysis of the as-cast ZM31-xY alloys.

**Figure 3 materials-13-00583-f003:**
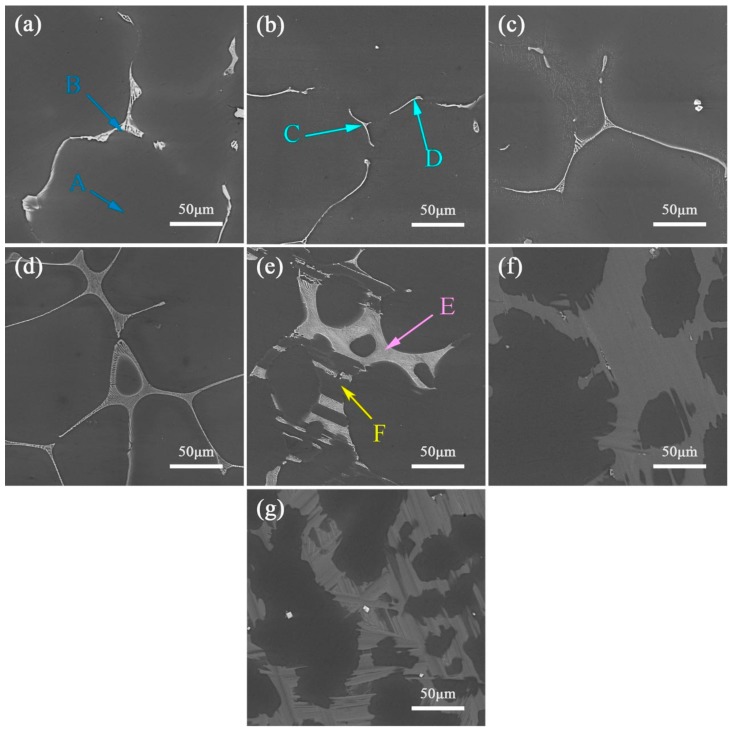
SEM micrographs of the as-cast ZM31-xY alloys: (**a**) x = 0; (**b**) x = 0.3; (**c**) x = 0.7; (**d**) x = 1.5; (**e**) x = 3; (**f**) x = 5; (**g**) x = 10.

**Figure 4 materials-13-00583-f004:**
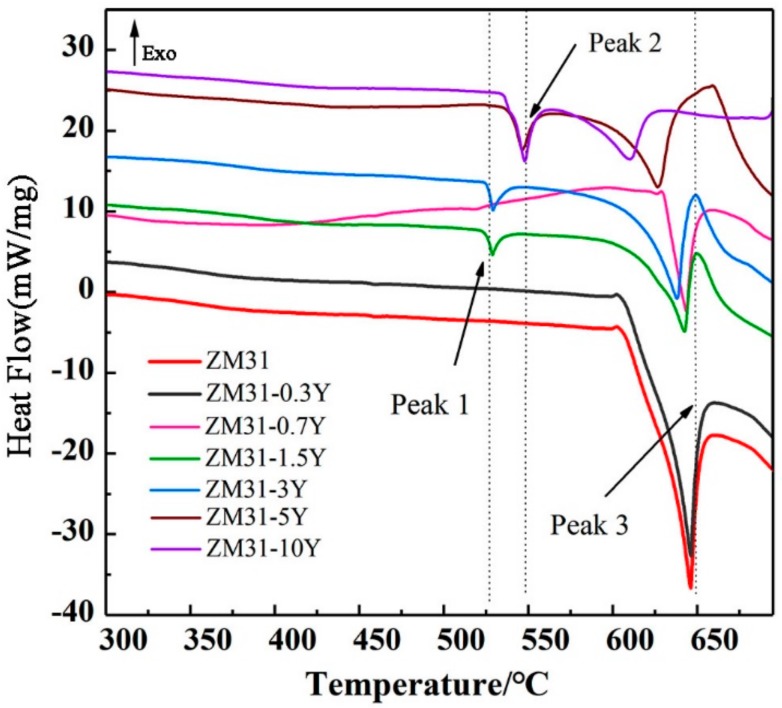
DSC curves of the as-cast ZM31-xY alloys.

**Figure 5 materials-13-00583-f005:**
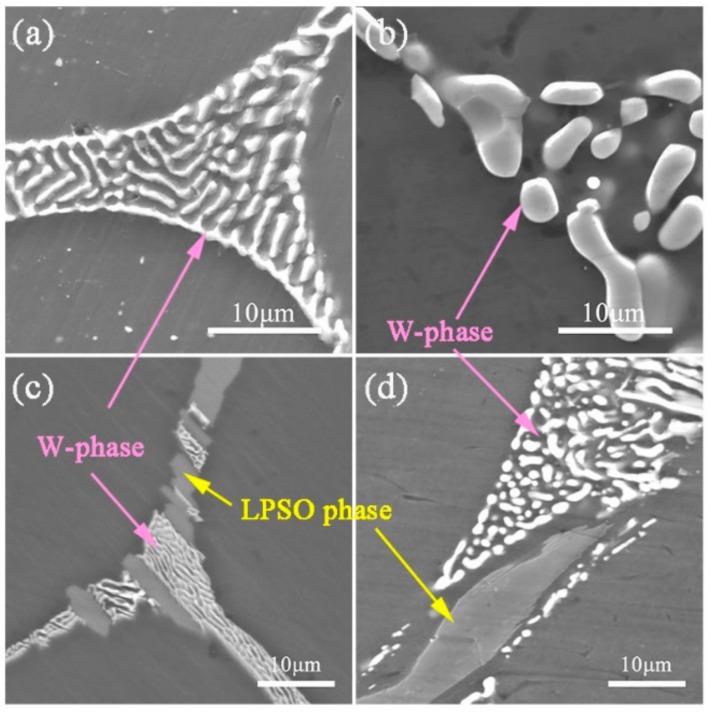
SEM micrographs of (**a** and **b**) ZM31-1.5Y and (**c** and **d**) ZM31-3Y alloys homogenized at (**a** and **c**) 420 °C/12 h and (**b** and **d**) 500 °C/16 h.

**Figure 6 materials-13-00583-f006:**
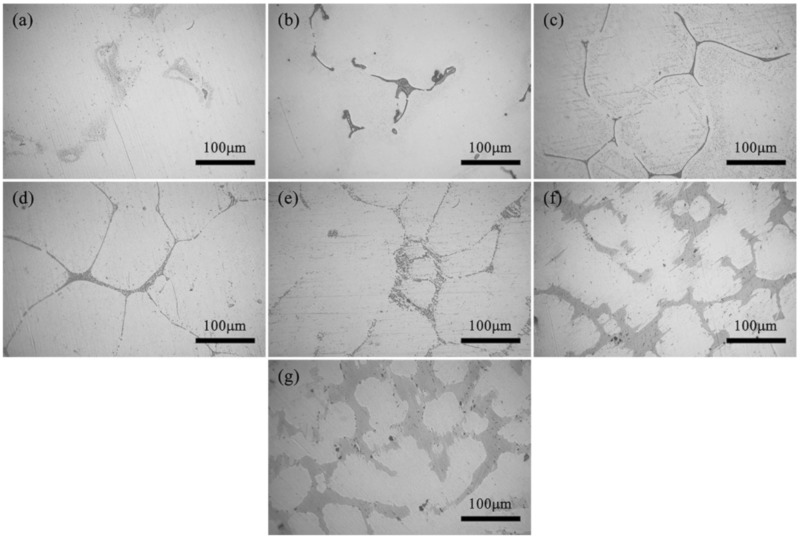
Optical micrographs of the as-homogenized ZM31-xY alloys: (**a**) x = 0; (**b**) x = 0.3; (**c**) x = 0.7; (**d**) x = 1.5; (**e**) x = 3; (**f**) x = 5; (**g**) x = 10. Homogenization at 420 °C/12 h: ZM31-(0, and 0.3) Y; Homogenization at 500 °C/16 h: ZM31-(0.7, 1.5, 3, 5 and 10) Y.

**Figure 7 materials-13-00583-f007:**
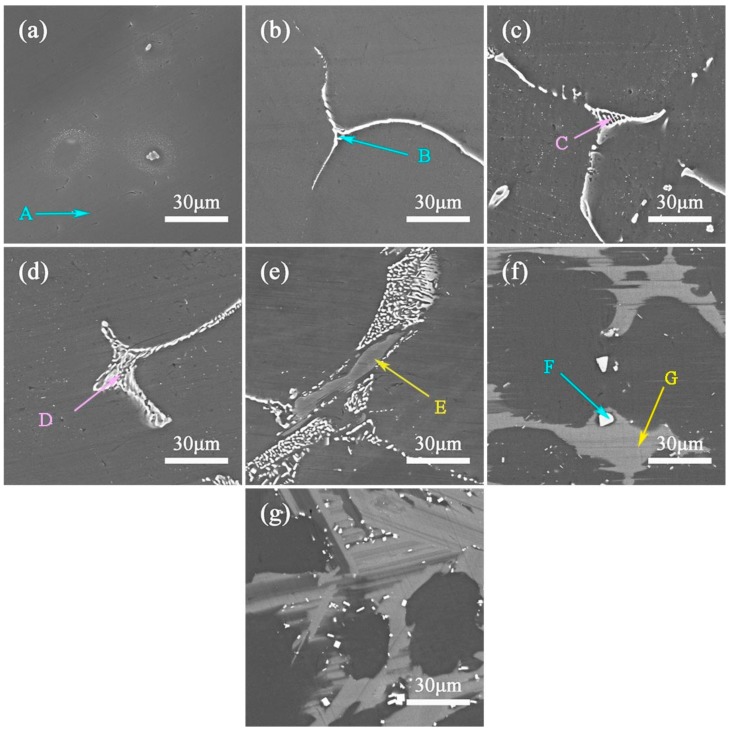
BSE-SEM micrographs of the as-homogenized ZM31-xY alloys: (**a**) x = 0; (**b**) x = 0.3; (**c**) x = 0.7; (**d**) x = 1.5; (**e**) x = 3; (**f**) x = 5; (**g**) x = 10.

**Figure 8 materials-13-00583-f008:**
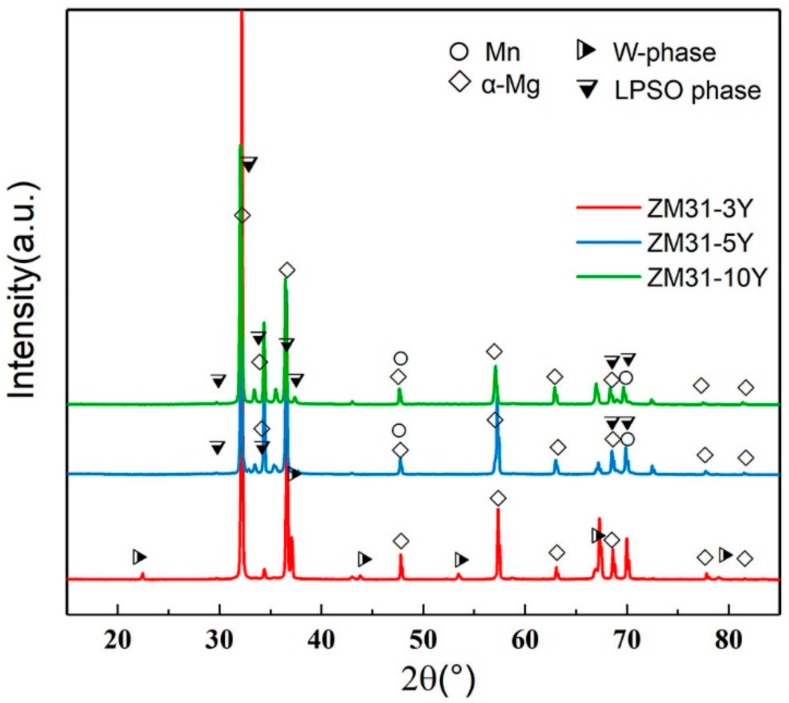
XRD patterns of the as-homogenized ZM31-xY alloys containing LPSO phase.

**Figure 9 materials-13-00583-f009:**
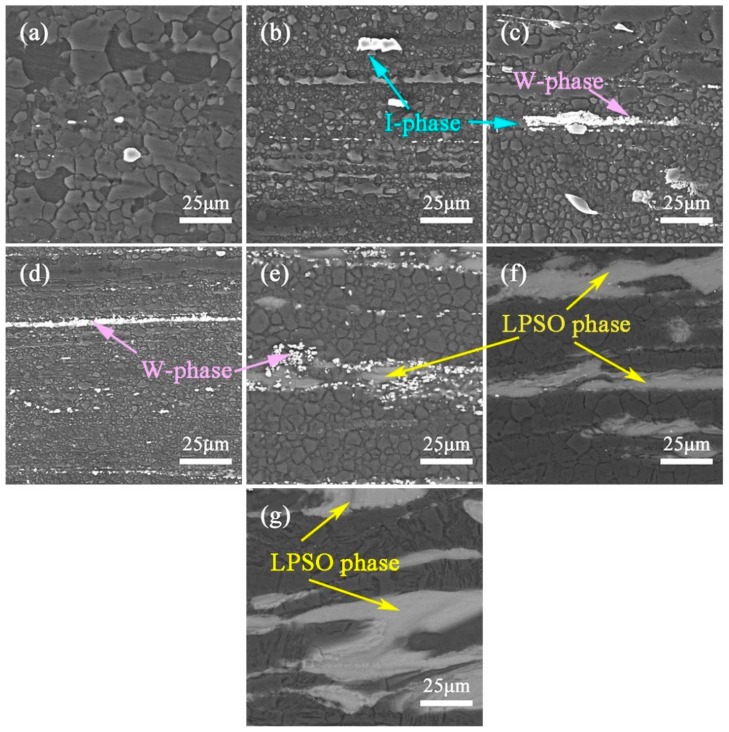
SEM micrographs of the as-extruded ZM31-xY alloys: (**a**) x = 0; (**b**) x = 0.3; (**c**) x = 0.7; (**d**) x = 1.5; (**e**) x = 3; (**f**) x = 5; (**g**) x = 10. At an extrusion temperature of 350 °C: ZM31-(0, 0.3, 0.7and1.5) Y; 480 °C extrusion temperature: ZM31-(3, 5and10) Y.

**Figure 10 materials-13-00583-f010:**
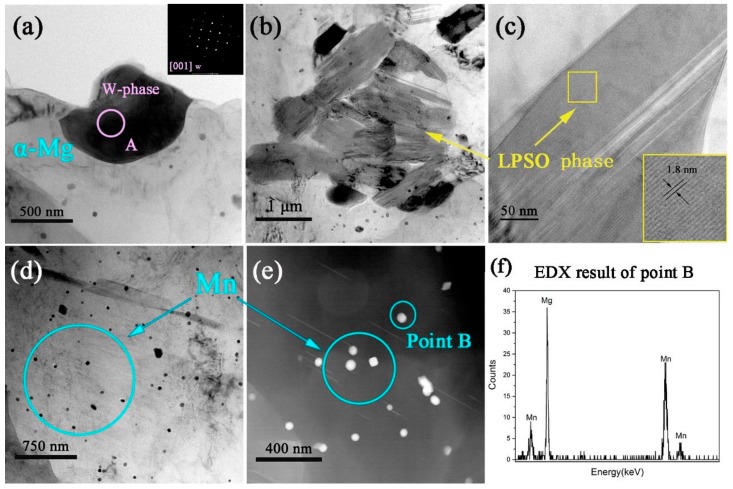
TEM micrographs of ZM31-3Y alloys: (**a**) BF-TEM image and selected area diffraction of the W-phase; (**b**) BF-TEM and (**c**) HR-TEM images of the LPSO phase; (**d**) BF-TEM and (**e**) HAADF-STEM images of Mn; (**f**) corresponding EDX results of point B in (**e**).

**Figure 11 materials-13-00583-f011:**
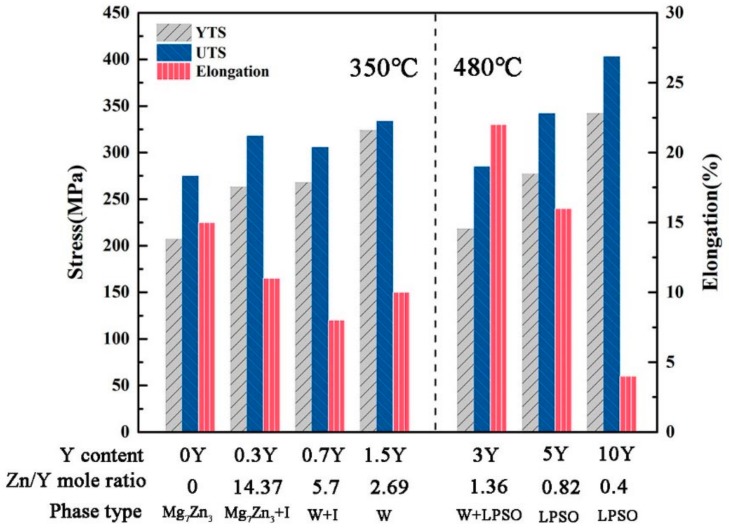
Mechanical properties of the as-extruded ZM31-xY alloys. 350 °C extrusion temperature: ZM31-(0, 0.3, 0.7 and 1.5) Y; 480 °C extrusion temperature: ZM31-(3, 5 and 10) Y.

**Table 1 materials-13-00583-t001:** The chemical compositions of the test alloys.

Nominal Alloys	Actual Composition (wt.%)			
	Mg	Zn	Mn	Y
ZM31	Bal.	3.00	1.08	-
ZM31-0.3Y	Bal.	2.74	0.91	0.34
ZM31-0.7Y	Bal.	3.15	0.92	0.64
ZM31-1.5Y	Bal.	2.56	0.79	1.47
ZM31-3Y	Bal.	2.61	0.76	2.83
ZM31-5Y	Bal.	2.93	0.79	4.83
ZM31-10Y	Bal.	3.24	0.83	10.93

**Table 2 materials-13-00583-t002:** Extrusion parameters for the studied alloys.

Test Materials	Billet Temperature (°C)	Extrusion Chamber Temperature (°C)	Mold Hole Diameter (mm)	Extrusion Ratio	Cooling Method
ZM31-xY (x = 0, 0.3, 0.7 and 1.5)	350	350	16	25	Air cooling
ZM31-xY (x = 3, 5 and 10)	480	480	16	25	Air cooling

**Table 3 materials-13-00583-t003:** Summary of main phases for the as-cast ZM31-xY alloys.

Nominal Alloys	Mole (Zn)/(Y)	Main Phases
ZM31-0Y	-	α-Mg, Mn and Mg_7_Zn_3_
ZM31-0.3Y	14.37	α-Mg, Mn, I-phase and Mg_7_Zn_3_
ZM31-0.7Y	5.70	α-Mg, Mn, I-phase and W-phase
ZM31-1.5Y	2.69	α-Mg, Mn, W-phase
ZM31-3Y	1.36	α-Mg, Mn, W-phase and LPSO phase
ZM31-5Y	0.82	α-Mg, Mn, W-phase and LPSO phase
ZM31-10Y	0.40	α-Mg, Mn and LPSO phase

**Table 4 materials-13-00583-t004:** EDS results of the as-cast ZM31-xY alloys in Figure 3.

No.	Mg (at.%)	Mn (at.%)	Zn (at.%)	Y (at.%)	Phase
A	97.9 (±0.11)	0.4(±0.15)	1.7 (±0.13)	0	α-Mg
B	72.4 (±0.13)	0	27.6(±0.21)	0	Mg_7_Zn_3_
C	47.5 (±0.09)	1.8 (±0.14)	42.4 (±0.44)	8.3 (±0.71)	I-phase
D	87.3 (±0.12)	0.4 (±0.13)	11.0 (±0.14)	1.7 (±0.28)	I-phase and Mg_7_Zn_3_
E	72.3 (±0.08)	0	16.4 (±0.83)	11.3 (±0.75)	W-phase
F	88.1 (±0.18)	0.7 (±0.32)	4.3 (±0.57)	6.9 (±0.23)	LPSO phase

**Table 5 materials-13-00583-t005:** Data of the DSC peaks in Figure 4.

Nominal Alloys	Peak 1	Peak 2	Peak 3
	Tp/°C	Tp/°C	Tp/°C
ZM31	-	-	646.24
ZM31-0.3Y	-	-	645.93
ZM31-0.7Y	517.31	-	643.54
ZM31-1.5Y	528.89	-	642.38
ZM31-3Y	529.07	-	638.33
ZM31-5Y	-	546.67	626.71
ZM31-10Y	-	547.81	609.55

**Table 6 materials-13-00583-t006:** EDS results of the as-homogenized ZM31-xY alloys in Figure 7.

No.	Mg (at.%)	Mn (at.%)	Y (at.%)	Zn (at.%)	phase
A	98.2 (±0.12)	0.5 (± 0.23)	-	1.3 (±0.13)	α-Mg
B	47.5 (±0.23)	1.9 (± 0.52)	8.2 (±0.63)	42.4 (±0.38)	I-phase
C	51.8 (±0.19)	0	20.3 (±0.43)	28.9 (±0.26)	W-phase
D	81.4 (±0.15)	0.4 (±0.15)	7.5 (±0.44)	11.7 (±0.27)	W-phase
E	88.1 (±0.23)	0.3 (±0.12)	6.7 (±0.45)	5.9 (±0.43)	LPSO phase
F	4.6 (±0.31)	-	93.2 (±0.16)	2.2 (±0.12)	Y
G	88.1 (±0.21)	-	6.8 (±0.44)	5.1 (±0.32)	LPSO phase

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
