# Peer review of "Effect of Y Addition on the Microstructure and Mechanical Properties of ZM31 Alloy"

_materials, 2020, doi:10.3390/ma13030583_

Round 1

Reviewer 1 Report

The authors suggest a very interesting analysis of different magnesium alloys of a ZM31 base alloy with increasing content of Y as alloying element and refer to the formation of different types of second phases in the as-cast and homogenized condition as well as their impact on the microstructure and mechanical properties during extrusion. The work is very comprehensive and offers important information for this highly relevant magnesium alloy system and is therefore of interest for the readers of the journal.

I suggest the acceptance of this paper for publication after a minor revision along the following suggestions:

I think the English needs a professional check – maybe the editorial check before publication will serve well. Lines 68 – 72: Can the authors please provide more detailed information why different annealing temperatures and times were chosen and also why different extrusion temperatures were selected. Especially for the extrusion conditions this selection hinders a comprehensive investigation of microstructure and property development with increasing Y content. Lines 198 ff: Can the authors please provide the average grain sizes of the alloys with higher Y content. It is understood that the grain structures will be coarser grained anyway. Lines 222: The authors would do well by also considering a Hall Petch type grain size effect on the mechanical behavior, not only the impact of the second phase. It can be assumed that the differences in the generally very fine grained materials have a significant impact on the yield behavior.

Reviewer 2 Report

This manuscript is dedicated to studying the influence of Y addition and morphology of Mg-Y-Zn phases on the mechanical properties of the ZM31-xY alloys. In my opinion the manuscript is original and provides interesting results, however there are a lot of errors and shortcomings. I recommend publishing this manuscript after major corrections given below.

1)   The authors examined the ZM31-xY alloys with the different amount of Y (x=0, 0.3, 0.7, 1.5, 3, 5 and 10%). Please explain why different temperatures of homogenization were used for the alloys: 420 °C for the ZM31 and ZM31-0.3Y alloys and 500 °C for the other alloys?

2)   What is the error of the measurement of chemical composition by means of the X-ray  fluorescence spectrometer (table 1)? In my opinion, the measurement accuracy should contain one decimal digit after comma. For example, 1.5±(?)% instead 1.47%.

3)   There is no description of the microstructure shown in Fig.1g in the text of the manuscript. 

4)   The authors wrote “the secondary phases on the grain boundary increase with the increase of Y content” (lines 118-119). Does it mean that the size or the volume fraction of the secondary phase particles increased?

5)   The EDS results presented in the table 3 are not correct:

a) the sum of all elements in the alloy should be 100%, while it is 102.71% in the measurement A, and 99,99% in the measurements D and F. Please correct it;

b) the correct result should contain one decimal digit after comma (not two). Please correct it and provide  the error of the measurement. For example, the quantitative analysis accuracy for the standardless analysis is 2% for the results of 100-20wt.%, 4% for the results of 20-5wt.%, 10-20% for the results of 5-1wt.% and 50%(up to 100%) for the results of 1-0.2wt.%;

c) please explain, why the I-phase (Mg3Zn6Y)  contains so wide amounts of Mg (47,6 – 87,2 at.%), Zn (42,4 – 11.0 at.%) and Y (8.1 –  1.4 at.%)? I think, that such spread of data and the mismatch in the chemical composition is the result of the small sizes of this phase. In other words, the sizes of this phase are too small for the correct measurement of the chemical composition by this EDS/SEM method. I recommend to perform the analysis of chemical composition in transmission electron microscopy in this case.

d) if the content of the individual elements changes in the wide range, please write in the table 3 the theoretical content range (according to the phase diagram) in the atomic percentage for the individual elements of the examined phases and the references of this data.

6) The chapter 3.2 concerns the descriptions of the microstructure of the as-homogenized ZM31-xY alloys. However, the first two paragraphs (lines 132-150) of this chapter describe the DSC result of the as-cast alloys. I suggest moving these two paragraphs to the chapter 3.1 and modifying its title.

7) The graph of the DSC curves is not complete (Fig. 4). Please mark what the ordinate axis means and mark the direction of the exo or endo reaction with the arrow.

8) The authors wrote that “From the previously analyzed SEM results, it can be seen that with the volume fraction of Y content increases, the amount of LPSO phase increases” (lines 168-169). I believe that only SEM results is not enough for this statement. Please present the XRD patterns of the as-homogenized alloys in order to compare them with the XRD patterns of the as-cast alloys.

9) Please correct the table 5: write only one decimal digit after comma and the error of the measurement.

10) Analyzing the SEM images of the as-extruded ZM31-xY alloys, the authors wrote “when the Y content is increased to 3%, the grain size of the alloy becomes large, which is mainly due to the appearance of LPSO phase” (lines 194-195). Since alloys with the Y content less than 3% were deformed at a lower temperature than alloys with a higher Y content, the question is: can the deformation temperature affect the grain size? The authors presented the grain sizes for the as-extruded alloys with content of 0.3, 0.7 and 1.5 at.% Y. What are the grain sizes for the ZM31 alloy after extrusion and as well as for the as-extruded alloys with 3, 5 and 10 at.% Y?

11) By means of TEM study the authors showed the present of the small Mn particles in the extruded ZM31-3Y alloy, that “can signifinantly refine the grain size”. What about the present of these particles in the extruded ZM31-5Y and ZM31-10Y alloys?

12) Why the authors wrote nothing about the changes of the elongation in the chapter of mechanical properties (chapter 3.4)? Please explain, why elongation significantly increased in the alloy with 3 at.%Y in comparison with the alloy with 1.5 at.%Y and why the elongation next decreased with the increase of the Y content? 

13) Please correct the conclusions:

a) The authors wrote that “The addition of Y can significantly refine the grain size of ZM31 alloy, which greatly improves the mechanical properties of the test alloys” (lines 252-253). However, according to the Fig.9, the best mechanical properties are observed in the alloys with 3, 5 and 10 at.%Y, while according to the Fig. 7, the grain sizes of the ZM31 alloy and alloys with 5 and 10 at.%Y are almost the same.

b) I think, that the sentence “The W-phase and LPSO phase have thermal stability property” (lines 256-257) is not correct, because generally all phases have thermal stability property. The authors did not detect an endothermic peak from the dissolution of the I-phase because of the small amount of this phase, but this does not mean that the I-phase has not thermal stability property. According to the DSC results, it would be correct to write that the thermal stability of the LPSO-phase is slightly greater than that of the W-phase.

c) I believe that the statement “the dispersed Mn phases can improve the mechanical properties of the test alloys and accelerate the formation of LPSO phase” (lines 259-260) is not enough proven at this work. The present of the Mn particles was showed only in the one as-extruded ZM31-3Y alloy.  

Round 2

Reviewer 2 Report

In my opinion the article is well corrected, the changes made it more clear  to understand. The author's responses to reviewers' comments are satisfying. I have no questions or comments and recommend publishing this paper in Materials.